# Stress Control in Older People through Healing Garden Activities

**DOI:** 10.3390/bs14030234

**Published:** 2024-03-13

**Authors:** Sun-Hee Kim, Joo-Bong Seo, Byung-Yeol Ryu

**Affiliations:** 1Graduate School of Environment Horticulture, Sahmyook University, Seoul 01795, Republic of Korea; sjb3325@hanmail.net; 2Department of Environment Horticulture, Sahmyook University, Seoul 01795, Republic of Korea; ryuby@syu.ac.kr

**Keywords:** cultivation management, mental, physical, the significance of the garden, utilization

## Abstract

This study, conducted on a group older than 60-years-of-age, sought to verify if healing garden activities control stress in older people. The experimental group performed garden activities once a week for 12 weeks, for 2 h each day, and the control group continued their daily lives. Each group’s cumulative stress at the beginning and end, along with total power (TP), the standard deviation of the normal-to-normal interval (SDNN), and the root mean square differences of successive R–R intervals (RMSSD), were measured using u-Bio MACPA, a stress index meter. The analysis showed that the experimental group had a statistically significant decrease in cumulative stress, compared to the control group; and also that TP significantly increased, compared to the control group. The SDNN and RMSSD of the experimental group increased, and decreased in the control group, but the changes were not statistically significant. The fact that cumulative stress decreased and stress evaluation indicators increased shows that daily stress can be controlled through healing garden activities. However, due to the small number of participants in the experiment, its ability to be generalized to all elderly people is subject to a number of limitations. Nevertheless, I think it is meaningful that the finding that garden activities are significant in mediating stress in the elderly was verified using a scientific measurement instrument. Future studies should explore the healing effectiveness of gardens in other age groups.

## 1. Introduction

With the improvement in living standards and the development of medical technology, the worldwide elderly population is rapidly increasing. In the case of Korea, the transition to an aging society is progressing more rapidly, and the country is expected to become a super-aged society, with 24.3% of the population being aged 65-years-of-age and older by 2030. The increase in the elderly population and the extension of life expectancy can cause various problems, such as economic poverty, physical aging, and loss of social status, which depend on the individual. These issues can sometimes spread into social problems. According to data from the National Statistical Office, 78.1% of seniors in their 60s and 79.0% of seniors in their 70s responded that they feel stress in their overall lives [1].

Stress is a process of reacting to events and environmental characteristics that are challenging, burdensome, or threaten happiness [2]. Stress can be brief, situational, and a positive force motivating performance; however, if experienced over an extended period it can become chronic stress, which negatively impacts health and well-being [3]. Mentally, it increases feelings of anxiety, loneliness, and depression; physically, it disrupts sleep, increases heart rate and blood pressure, reduces the body’s ability to make antibodies, weakens the immune system, and slows the healing of wounds. Stress is harmful by itself, and can also produce negative health effects such as heart disease, cancer, and diabetes, and lead to health behavioral problems, such as drinking and smoking, irregular eating, and negative exercise habits [2]. Although everyone experiences stress because it is a normal part of life, the intensity, frequency, and duration of stress can vary from person to person [4]. For older people, the stress experienced in daily life has a stronger emotional and psychological impact than major life events [5]. In the elderly, stress mediation is more important because, although people experience more stress as they age, their ability to find effective ways to relieve it decreases [6]. Research of the occurrence and resolution of stress in the elderly has been continuously conducted, and the need for various programs regarding stress management methods has also been emphasized [7].

For people in modern society who live with a lot of stress, gardens are a space of relaxation and recovery. Healing gardens are designed to consider therapeutic design elements such as the sounds, sights, and smells of the environment that help participants regain health and improve well-being [8]. Scent particles from various plants in the garden contact the nasal mucosa and reach the limbic system of the brain via the olfactory nerve, thereby strengthening cerebral mental function, reducing the stress response [9], improving immune function, and leading to changes in response [10]. In addition, a healing garden is a place to recover, maintain, and improve physical and mental health, as it encourages physical exercise. It is also a space that can aid in cognitive recovery through gardening, an act of directly experiencing nature [11]. Healing garden activities allow participants to use their muscles when grasping tools (compacting the soil with a shovel or planting plants) and are effective in exercising both the small and large muscles (improving the flexibility of the thumb and index finger) and, as such [12], lead to improved physical abilities. In addition, healing garden activities are known to be effective in the recovery and maintenance of physical and mental health because one can experience the intellectual effect of acquiring new knowledge and utilizing technology, as well as the social effect of forming social relationships by interacting with others [13]. These garden benefits are especially important for older people. In elderly care facilities where elderly patients live, walking in the healing garden and communing with nature not only relieves stress but also provides positive effects that can improve various physical activities and perceptual abilities [14].

This study aims to confirm that gardens can be used as a tool to maintain and improve the physical and emotional health of older people in the wider context of an ageing population. We aim to identify stress reduction and physical changes in older adults when they are introduced to a gardening activity program that increases physical activity and provides psychological and emotional relief.

## 2. Materials and Methods

### 2.1. Participants

The subjects in the study were Seoul citizens over 60-years-old, who are residents of Eunpyeong-gu, and able to freely move their bodies and express their opinions. The control group consists of users of the oo senior center located in Eunpyeong-gu, who are seniors over 60-years-of-age who, after indicating their intention to participate, freely expressed their opinions.

### 2.2. Experimental Procedures

The 20 participants in this experiment were recruited through the Eunpyeong-gu Office integrated notification system and the Hyanglim Urban Agricultural Research Institute website. A detailed description of the study was posted in the recruitment announcement. The control group consisted of 20 elderly people, who expressed their willingness to participate in the study. Of the 20 people in each group, 19 people in each group participated in the experiment, excluding the 1 person in each group who did not meet the conditions. The experiment was conducted for 12 weeks in the period from 22 April to 1 July 2023. In being based on previous research, the program was organized once a week, with a total of 12 sessions. This was done to reduce any decline in physical function and decrease emotional dissatisfaction, which are causes of stress in the elderly, and to improve exercise muscle function and achieve physical and physiological balance effects.

Participants’ cumulative stress index and heart rate variability (HRV) were measured before participation and at 4, 8, and 12 weeks after participation. The detailed items of HRV include immunity index (total power (TP)), stress resistance index (standard deviation of NN intervals (SDNN)), and physiological health status (square root of the mean of the sum of the square of differences between adjacent NN intervals (RMSSD)), which are each items that analyze cumulative stress. Among the subjects, 11 participants in the experimental group and 11 in the control group were selected as subjects for the final analysis, excluding 8 participants who had a low program rate of absence and 8 participants who did not participate in the measurements (Figure 1) from across both groups.

#### 2.2.1. Healing Garden Activity Program

The healing garden activity program allows people to experience a variety of activities such as planting, cultivating and managing plants, creating a garden and utilizing harvested produce; along with meditating and exercising in nature, and taking walks. Garden creation and planting activities included creating a garden, sowing and planting herbs and edible flowers, setting and planting pet plants, creating a mini garden, and growing vegetable sprouts, as well as regular cultivation and maintenance of the garden, such as watering, thinning, fertilizing, and propagation. A “farm party”, which is a community activity centered on the garden, was also held, and included activities such as drinking herbal tea using harvested plants, making pressed flowers, flower arranging, natural dyeing, and making natural object frames (Table 1).

#### 2.2.2. Measuring Tool

In this study, a portable stress measurement machine, u-Bio Macpa (u-Bio Clip v70, Biosense Creative Co., Ltd., Seoul, Republic of Korea) was used, instead of the commonly used questionnaire, to measure stress on a scientific basis. To determine the state of stress, heart rate variability (HRV) was measured. Methods for analyzing HRV include the time domain analysis and frequency domain analysis [15]. u-Bio Macpa waves were analyzed. By connecting the fingertip to a portable autonomic nervous system measurement clip, infrared rays coming from the direction of the fingertip passed through the finger blood vessels, and blood vessel health and cumulative stress index were measured through a signal analysis of pulse waves measured for 5 min [4,16]. Sympathetic nerve activity (low-frequency), parasympathetic nerve activity (high-frequency), autonomic nerve balance (LF/HF ratio), and total power (TP) were measured using the frequency domain analysis method; body resistance (standard deviation of NN intervals (SDNN)) was measured using the time domain analysis method; and square root of the mean of the sum of the square of differences between adjacent NN intervals (RMSSD) was used to analyze the stress accumulated in the body. TP provides an evaluation of the overall autonomic nervous system activity, as well as sympathetic nerve activity with total power, including VLF, LF, and HF power measured over 5 min [15]. Lower values usually indicate being too tired or sick. Higher TP is better but should not be too high [17]. Total power in the frequency domain has a similar meaning to SDNN in the time domain [18]. SDNN is a significant indicator of short-term variation in heart rate in short-term (at least 5 min) heart rate variability analyses, and shows if the autonomic nervous system can control the body [19]. A decrease in SDNN results in a loss of the ability to cope with various stresses and a decline in overall health. On the other hand, RMSSD shows long-term variations in heart rate in heart rate variability analyses and measures the activity of the parasympathetic nervous system, and can therefore evaluate the control ability of the parasympathetic nervous system [18]. u-Bio Macpa sums the values of each item in the detailed analysis and expresses them numerically, presenting a score from 0 to 100 (Patent No. 10-0954817).

### 2.3. Statistical Analysis

Stress measurement and HRV analysis using u-Bio Macpa are based on the standards of the European Society of Cardiology and the North American Society of Cardiac Pacing and Electrophysiology, as well as the guidelines for autonomic nervous system evaluation and interpretation, and analyzed by using Biosense Creative (Seoul, Republic of Korea) for u-Bio Clip v70 [19]. A state of almost no stress is 25 or less; temporary stress is represented by a score from 26 to 35; initial stress is represented by a score from 36 to 45; a state where tolerance to stress begins to weaken is represented by a score from 46 to 59; and chronic stress is above 60. The higher the score, the more stressful the event, which means that the index is high. The reference values of TP, SDNN, and RMSSD vary depending on age. In the age range from 50 to 70 years-of-age, the reference ranges are from 6.96 to 8.87 Ln for TP, from 20 to 100 bpm for SDNN, and from 10 to 60 bpm for RMSSD. Within this range, it has been reported that the higher the number, the healthier the stress levels [20].

The effectiveness of this program was analyzed using a non-parametric test because the number of participants in each group did not exceed 30 [4]. Pre- and post-average comparisons within groups for healing garden activities were analyzed using the Wilcoxon signed-rank test, and mean comparisons between experimental and control groups were conducted using the Mann–Whitney U-test. All statistical processing was performed using SPSS version 29.0 (IBM Co., Chicago, IL, USA), and all statistical significance levels for the data were set at *p* < 0.05.

## 3. Results

The sexual distribution of the experimental group and the control group (men 27.3%) were the same. All participants in the control group and almost all participants in the experimental group were married. Regarding the distribution of age, the experimental group has a higher percent of participants aged 60–69 years (72.7% vs. 54.5%) and a lower percent of participants aged 70–79 years-of-age (27.3% vs. 45.5%). In terms of educational background, the experimental group has a lower percentage of high school graduates than the control group (45.5% vs. 81.8%). In terms of work experience, the experimental group had equally high proportions of those between 8 to 12 years and more than 21 years, but the control group had the highest proportion of those between 8 to 12 years (27. 3% vs. 54.5%) (Table 2).

### 3.1. Test for Homogeneity between Gardening Group and Control Group

The Mann–Whitney U-test was performed as a preliminary measurement to determine if the two groups were physiologically homogeneous. The stress index, TP, SDNN, and RMSSD were all *p* > 0.05, showing no significance between the two groups, confirming they are homogeneous groups (Table 3).

### 3.2. Changes in Stress and TP, SDNN, and RMSSD of the Gardening Group

Table 4 shows the degree of change in the experimental group’s physical responses as garden activities progressed. Cumulative stress levels steadily increased during the garden activities, but dropped sharply in the 12th week, decreasing by an average of 3.82 from the initial 56.26 to 52.44. Total power (TP) steadily increased from the start of the program, increasing by an average of 0.25. The standard deviation of the normal-to-normal interval (SDNN) decreased at Week 4, then began to increase from Week 8, before then increasing sharply at Week 12. There was an average increase of 2.78, compared to the baseline. Root means square differences of successful R–R intervals (RMSSD) also showed the same changes as SDNN. They decreased in the 4th week, but increased starting from the 8th week, showing an average increase of 1.34 in the 12th week, compared to the baseline. These dependent variables were not found to be statistically significant. However, when Week 0 and Week 12 were compared, stress decreased overall and TP, SDNN, and RMSSD values showed an increasing trend (Figure 2).

### 3.3. Changes after the Program in the Gardening Group and Control Group

Comparing the (pre and post) values of the experimental and control groups, the stress index of the experimental group are seen to have decreased by approximately 6.8%, while the stress index of the control group increased by approximately 6.5%, showing an average increase of 3.41. The difference was statistically significant (Figure 3).

The TP level showed an increase of 0.25 Ln in the experimental group, while the control group showed a decrease of 0.27 Ln. The value was statistically significant. For SDNN, the experimental group showed an increase of approximately 2.78, while the control group showed a decrease of 0.19. RMSSD also increased by 1.34 in the experimental group, but also increased by 1.67 in the control group, meaning this was not a significant result (Table 5).

## 4. Discussion

The purpose of this study is to investigate if r healing garden activities can control stress in elderly people. The results of the study showed that the cumulative stress level of the elderly population who participated in the healing garden activities decreased by 6.5%, compared to the control group who continued their daily lives. In other words, regular physical activity releases stress and decreases daily stress levels [21]. Cultivating and managing garden activities, such as watering, sowing, weeding, pruning, and harvesting, are found to, when performed daily in the garden, require continuity and regularity, in the sense of cyclic seasonal changes and the repeated act of nurturing. In addition, cultivating and maintenance garden activities mostly utilize large muscles in outdoor spaces and help to increase body movement, and also improve motor function. Although seemingly easy and simple compared to daily exercise, garden activities are easy to start and are considered appropriate for increasing the physical activity of the elderly, who often have poor physical health and a lack of strength because of a lack of physical activity.

The effect is also clearly visible in the HRV results of elderly people who participated in healing garden activities. In the case of TP levels, the elderly population who participated in the healing garden activities showed an increase of 0.25, but the control group showed a decrease of 0.29, which outside the reference range. If TP is outside the reference range, this means that the immune function is also outside the normal range. Immune function affects the stress response. Stress stimulates the sympathetic nervous system to release hormones as a survival instinct. These hormones are produced by the sympathetic-adrenal-medullary (SAM) axis and the hypothalamic-pituitary-adrenal (HPA) axis. The immune system is directly disrupted by the endocrine response to stress. Immune function is linked to the autonomic nervous system and the pituitary-adrenocortical system, which influence the stress response [22]. When stressed, the body’s sympathetic nervous system is activated, blood cortisol levels increase, and immune function decreases [23]. These findings are consistent with previous research [24] that found that spending time in the garden reduced participant’s cortisol (a chemical the body produces in response to stress) levels more than reading a book. The healing garden activity program was considered to reduce stress and increase immune function in the elderly group who participated in the program. In contrast, the elderly population who continued their daily lives had no medium to mediate daily stress, with the result that stress accumulated and resulted in decreased TP levels, signifying decreased immune function.

SDNN reflects the response of the autonomic nervous system, which is involved in responding quickly and appropriately to changes in both the body and the external environment. The higher the SDNN, the healthier the person [25]. RMSSD is an index of heart-rate fluctuations, which reflects the parasympathetic state and is found to decrease overall in groups with high stress [26]. The SDNN and RMSSD of elderly participants in the healing garden activity increased by 2.78 and 1.34, respectively. In the control group, SDNN decreased by 0.19 and RMSSD increased by 1.67, which suggests that healing garden activities, like other stress intervention programs, help maintain a physiologically healthy state by improving the overall activity and control of the autonomic nervous system. Positive changes in participants’ HRV levels were considered to have been due to a sense of peace from being close to plants and being given an opportunity to escape stress, even for a moment [27].

Healing garden activities can also be expected to have emotional effects that are difficult to provide in physical activity-oriented programs, such as yoga, dance, and gate ball, which are widely applied in stress intervention for the elderly. Meditation, walking, and sharing thoughts in the garden are also suitable emotional coping methods that can be used to relieve stress by making small life changes or exchanging emotions [28]. Collaborative efforts to grow and share herbs, fruits, and vegetables can provide opportunities to gain a sense of accomplishment and improved self-esteem. Connecting with other members of the community through community activities, such as farm parties, can help participants to gain social support and contribute positively to the lives of others, which also helps to protect against isolation and loneliness [29]. The elderly group that engages in green activities for leisure showed lower stress levels, compared to other groups that engage in travel, gatherings, and exercise [23], which shows that healing garden activities control stress more effectively than other stress intervention programs. Anyone can easily participate in healing garden activities without any special skills or knowledge, and there are no spatial restrictions. Kaplan, in introducing the concept of “nearby nature”, stated that any type of activity, such as growing indoor plants, looking at trees through a window, tending a garden, or looking at trees or flowers on the street, contributes positively to controlling stress [30]. In other words, simply enjoying the space by using the senses could be considered effective in controlling stress. In addition, touching plants can reduce physiological and psychological stress by lowering blood pressure and heart rate, affecting the cardiovascular system through contact [31]; gardening, as an activity that induces continuous and long-term contact with plants, can be said to be highly effective in this regard.

Although this study confirmed that healing garden activities are effective in controlling stress in the elderly, the study also has limitations. First, there were only 38 participants in the experimental and control groups, making it difficult to generalize effectiveness to all elderly people. To compensate for this, the number of participants needs to be expanded, and further research of effectiveness for different ages will be needed. Additionally, due to the nature of gardens, where the environment changes with changes in the season, the effectiveness of garden activities during a hot summer or cold winter has not been confirmed. Considering this characteristic, research results will need to be obtained from mid- to long-term programs. In this study, only measuring instruments were used to obtain scientific results for the effect of stress reduction, which made it difficult to capture various aspects of stress. In order to achieve realistic and effective stress intervention, the use of psychological assessment tools should also be considered.

Currently, healing garden activities that target the elderly generally focus on special groups, such as those with dementia or depression. However, it is necessary to expand the scope of what is considered to be healing and to conduct research into the effectiveness of easily implementable healing gardens as a preventative measure, and to also assess the subsequent development of such programs. To achieve this, urban gardens should also be considered a potential space for healing activities in everyday life.

## Figures and Tables

**Figure 1 behavsci-14-00234-f001:**
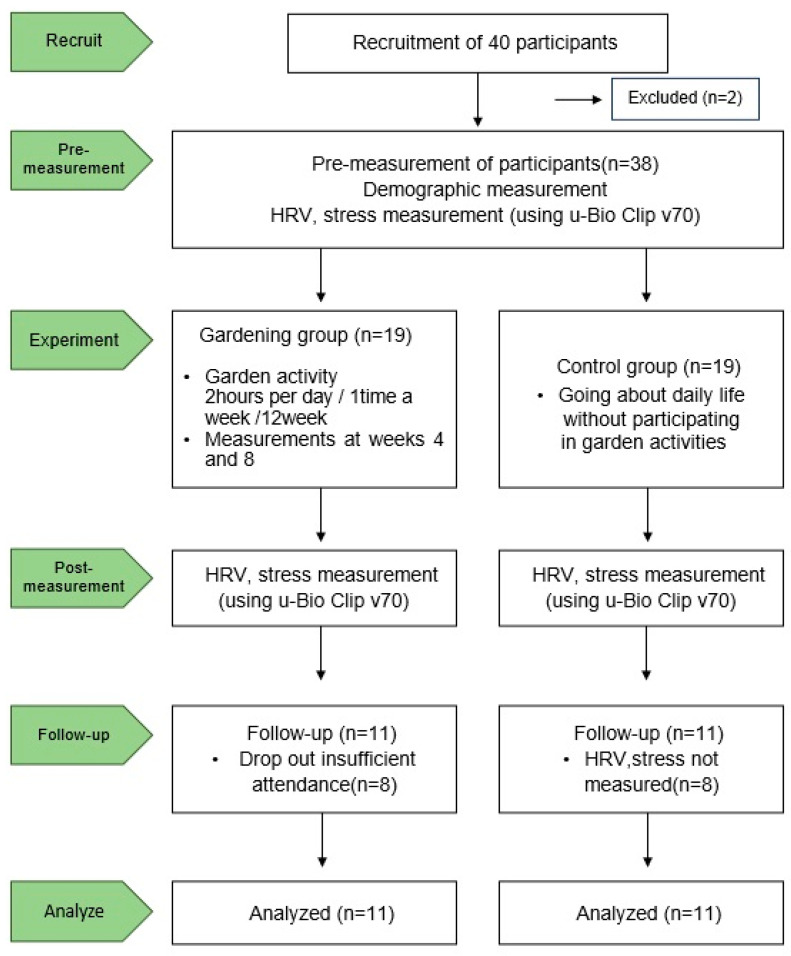
Flow diagram of total experimental procedure.

**Figure 2 behavsci-14-00234-f002:**
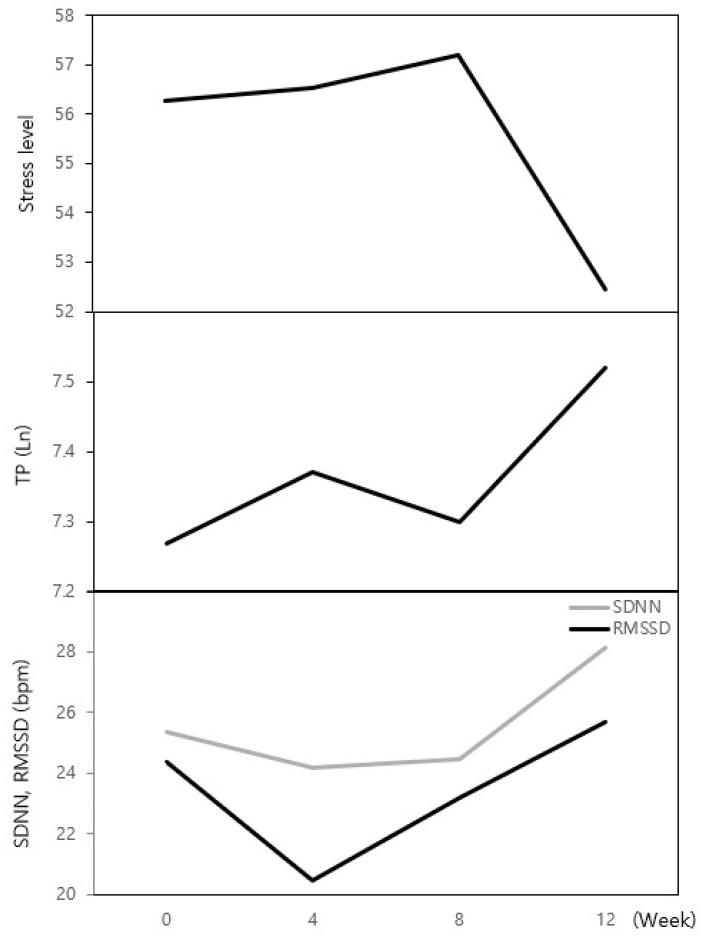
Changes in stress and the TP, SDNN, and RMSSD of the experimental group over time. Stress rose slightly at Week 4, reached its peak at Week 8, and decreased sharply at Week 12. TP increased from Week 4, decreased briefly at Week 8, and reached its highest level at Week 12. Although there are differences in the values of SDNN and RMSSD, they both tend to decrease in the 4th week and then gradually increase, before reaching their maximum level in the 12th week.

**Figure 3 behavsci-14-00234-f003:**
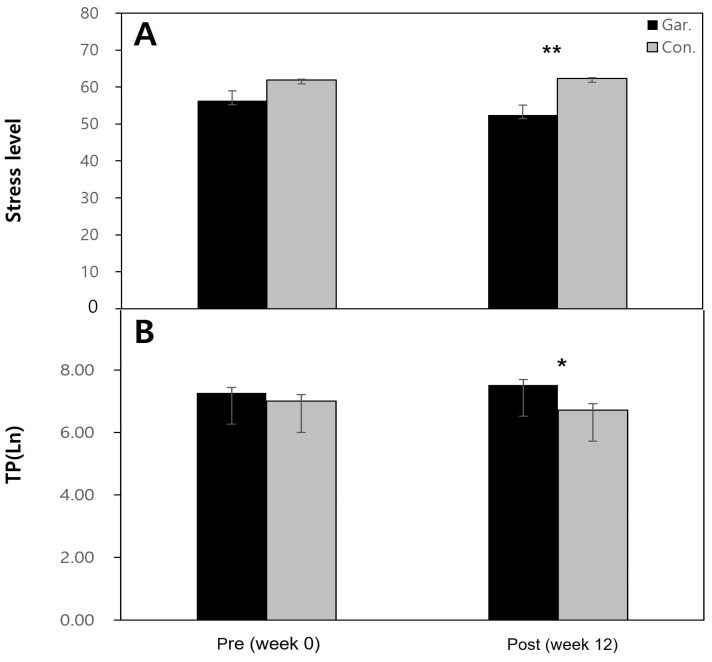
Changes in the stress level and TP of gardening participants and the control group after the gardening program: (**A**) After the gardening program, the stress level decreased in gardening participants, and increased in the control group. (**B**) After the gardening program, TP slightly increased in gardening participants, and decreased in the control group. * *p* < 0.05, ** *p* < 0.01).

**Table 1 behavsci-14-00234-t001:** Healing garden activity program.

Class	Topic	Garden Activities and Test for Stress	Healing Effect
1	We met on the green	Program orientation and pre-test (u-Bio) Introductions A tour of the garden	Rapport formation
2	A budding seed in my heart	Fertilizing the garden planting annuals A plant that resembles my heart	Improved athletic performance and cognitive ability.
3	Herbs and Flowers	Planting herbs and edible flowers Showing off my pet plants	Improved athletic performance, comfort, psychological stability, and self-esteem.
4	Beautiful, my life	Taking care of the garden and my heart Making a terrarium Week 4 u-Bio test	Improved activity and tactile ability
5	A scented day	Walking in the herb garden My favorite scent/making herb soap	Psychological stability; improved perception
6	Forest is my friend	Forest meditation Taking care of the garden	Recognition of seasonal and temporal changes; strengthening cardiopulmonary function
7	Flavor and coolness	Cooking with herbs and edible flowers Sharing food	The joy of harvest; reminiscing about the past
8	The growing number of green friends	Greenwood cutting Taking care of the garden Week 8 u-Bio test	Improved athletic performance and orientation.
9	I’m a flower	Cutting flowers in the garden Arranging flowers/showing my flowers	The joy of harvest; the sense of the season
10	The pigments of flowers	Dyeing scarves with natural dyes Talking about my color	Improved visual ability and change in perception.
11	Mini my garden	Making a succulent garden Taking care of the garden	Improved athletic performance, concentration, and orientation.
12	A heart-sharing garden	Taking care of the garden Creating a natural object frame Week 12 u-Bio test	Improved athletic performance and gratitude.

**Table 2 behavsci-14-00234-t002:** Demographic characteristics of the study population and the dependent variable homogeneity of the dependent variable.

Demographics		Gardening *N* (%)	Control *N* (%)	Χ^2^	*p*
Gender	Male	3 (27.3)	3 (27.3)	4.545	0.682
	Female	8 (72.7)	8 (72.7)		
Marital status	Unmarried	1 (9.1)	0 (0.0)	1.048	0.306
	Married	10 (90.9)	11 (100)		
Age (years old)	60 to 69	8 (72.7)	6 (54.5)	0.786	0.375
	70 to 79	3 (27.3)	5 (45.5)		
	80 to 89	0 (0.0)	0 (0.0)		
Education	High school graduate	5 (45.5)	9 (81.8)	5.143	0.162
	Collage graduate	4 (36.3)	0 (0.0)		
	University graduate	1 (9.1)	1 (9.1)		
	Master’s or higher degree	1 (9.1)	1 (9.1)		
Work Experience (years)	Less than 3	2 (18.1)	0 (0.0)	3.867	0.424
	4 to 7	1 (9.2)	2 (18.1)		
	8 to 12	3 (27.3)	6 (54.5)		
	12 to 20	2 (18.1)	1 (9.1)		
	More than 21	3 (27.3)	2 (18.1)		
Stress Level Indicator	Almost no stress (>25)	0 (0.0)	0 (0.0)	5.200	0.158
	Temporary stress (26 to 35)	1 (9.1)	0 (0.0)		
	Initial stress (36 to 45)	0 (0.0)	1 (9.1)		
	Tolerance stress (46 to 59)	7 (63.6)	3 (27.3)		
	Chronic stress (<60)	3 (27.3)	7 (63.6)		

**Table 3 behavsci-14-00234-t003:** Pre-homogeneity test of gardening group and control group.

Analysis	Gardening	Control	z	*p*
Stress Level	56.26 ± 10.54 ^w^	61.87 ± 10.55	−1.22	0.22 ^NS^
TP ^z^	7.27 ± 0.74	7.01 ± 0.84	−0.85	0.39 ^NS^
SDNN ^y^	25.34 ± 10.74	22.22 ±10.06	−0.69	0.49 ^NS^
RMSSD ^x^	24.36 ± 19.74	17.93 ± 9.93	−1.28	0.20 ^NS^

^z^ TP: Total power. ^y^ SDNN: The standard deviation of the normal-to-normal interval. ^x^ RMSSD: Root mean square differences of successive R–R intervals. ^w^ Mean ± Standard deviation (*N* = 11). ^NS^ Non-significant at *p* < 0.05 leveled by Mann–Whitney U-test.

**Table 4 behavsci-14-00234-t004:** Changes in stress and TP, SDNN, and RMSSD of the gardening group (*n* = 11).

Analysis	Pre (Week 0)	Week 4	Week 8	Post (Week 12)	F	*p*
	M ± SD	M ± SD	M ± SD	M ± SD		
Stress Level	56.26 ± 10.54 ^w^	56.53 ± 13.22	57.21 ± 9.71	52.44 ± 12.84	2.55	0.13 ^NS^
TP ^z^	7.27 ± 0.74	7.37 ± 0.65	7.30 ± 0.49	7.52 ± 0.57	0.49	0.70 ^NS^
SDNN ^y^	25.34 ± 10.74	24.17 ± 14.69	24.46 ± 7.17	28.12 ± 11.55	0.93	0.47 ^NS^
RMSSD ^x^	24.36 ± 19.74	20.46 ± 14.34	23.20 ± 10.54	25.70 ± 15.63	1.09	0.41 ^NS^

^z^ TP: Total power. ^y^ SDNN: The standard deviation of the normal-to-normal interval. ^x^ RMSSD: Root mean square differences of successive R–R intervals. ^w^ Mean separation within columns by ANOVA at *p* < 0.05. ^NS^, non-significant.

**Table 5 behavsci-14-00234-t005:** Post-program changes in the gardening and control group.

Analysis	Gardening	Control	z	*p*
Stress Level	52.44 ± 12.84 ^w^	65.28 ± 6.92	−2.66	0.008 **
TP ^z^	7.52 ± 0.57	6.72 ± 0.66	−2.30	0.022 *
SDNN ^y^	28.12 ± 11.54	22.01 ± 8.94	−1.02	0.31 ^NS^
RMSSD ^x^	25.70 ± 15.63	19.60 ± 12.37	−1.28	0.20 ^NS^

^z^ TP: Total power. ^y^ SDNN: The standard deviation of the normal-to-normal interval. ^x^ RMSSD: Root mean square differences of successive R–R intervals. ^w^ Mean ± standard deviation (*N* = 11). ^NS^, *, ** Found to be non-significant or significant at the *p* < 0.05 or 0.01 levels by the Mann–Whitney U-test, respectively.

## Data Availability

Data are contained within the article.

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
