# Peer review of "Stress Control in Older People through Healing Garden Activities"

_behavsci, 2024, doi:10.3390/bs14030234_

Round 1

Reviewer 1 Report

Comments and Suggestions for Authors

Introduction

P1 Line 28 “various social problems such as economic poverty, physical aging, and loss of social status”

   Economic poverty, physical aging, and loss of social status are individual problems, not social problems.

Line 31-47 This paragraph should be organized and shortened.

Line 65 Line 70  Sentences with the same content are repeated.

2 Materials and methods

Participants

P2 line 74-

The characteristics of participants is not written sufficiently.

No description about control group.

Experimental Procedure

Important information are lacked and unnecessary information is written.

Line 82-88  unnecessary

Information about how participants were recruited and about allocation are lacked.

Figure 1

What does “Everyday life” and “Lost to follow-up” mean? Correction of English is necessary.

P3 Total power is not common. Authors should explain the significance/meaning of this index and reference. How is this indicator measured?

P4 Line 122 Authors should specify the source of calculation of stress index.

Is it all right to add values of HRV, LF, HF, autonomic balance, SDNN, and RMSSD? Unit of each subindex are different.

Overall, the explanation for measuring indices is insufficient.

Results

Line 138 Looking at the general characteristics・・・ Such expression as “Looking at・・“is not used in scientific article.

Authors should describe the comparisons between experimental group and control group in demographics.

Table 3 Pre-homogeneity test?  In general, “test for homogeneity” is used

Discussion

Line 205-213 Authors should shorten these descriptions. The importance is discussion about results of this study,

Authors should discuss logically why gardening activity exerted healing effect using previous literatures.

Comments on the Quality of English Language

Authors should receive English editing service by native speaker. There are a lot of expressions that cannot be understand or too be difficult to understand because of poor  quality of English

Reviewer 2 Report

Comments and Suggestions for Authors

The article titled "Stress Control in Elderly through Healing Garden Activities" is structured into several sections, each focusing on different aspects of the research conducted. The importance of the article "Stress Control in Elderly through Healing Garden Activities" lies in its exploration of innovative, nature-based interventions for stress reduction among the elderly. This study adds to scientific literature by providing empirical evidence on the effectiveness of healing gardens as a therapeutic tool for the elderly, a demographic increasingly affected by stress-related issues. Its findings contribute to a growing body of research on environmental psychology and geriatric care, offering practical insights for healthcare providers, caregivers, and urban planners. By focusing on a non-pharmacological approach to stress management in the elderly, the article underscores the significance of incorporating natural environments into aging-related healthcare and wellness strategies.

Here's a detailed review report with highlighted areas of improvement for each section:

1. Abstract: The abstract concisely summarizes the study, its methodology, and findings. However, it could benefit from a brief mention of the study's limitations and potential implications for future research to provide a more comprehensive overview.

2. Introduction: This section effectively sets the context for the study by discussing the increasing elderly population and associated stress factors. To improve, it could include more recent statistics and a clearer linkage between stress in the elderly and the specific benefits of healing garden activities.

3. Materials and Methods:

    - Participants: This subsection is clear in describing the participant selection criteria. Including a more detailed demographic breakdown could enhance the understanding of the sample's representativeness.

    - Experimental Procedures: The description of the procedures is detailed. It could be improved by elaborating on how the activities were specifically designed to reduce stress.

    - Measuring Tool: The tool used for measuring stress is adequately described. Adding a comparison with other stress measurement tools might provide better insight into its effectiveness.

4. Statistical Analysis: The methods used for statistical analysis are appropriately detailed. It would be beneficial to discuss the choice of non-parametric tests over other statistical methods to clarify their appropriateness for the data.

5. Results: This section is well-organized and presents the findings clearly. Including visual aids, such as graphs or charts, could enhance the readability and immediate comprehension of the results.

6. Discussion: The discussion insightfully interprets the results, linking them back to the hypothesis and previous research. It could be strengthened by addressing potential confounding variables and discussing the results in the context of broader societal impacts.

7. Conclusions: The concluding section effectively summarizes the study's findings and their significance. Enhancing this section with recommendations for practical applications of the research and suggestions for future studies could provide a more impactful conclusion.

8. References: The references are comprehensive. Ensuring that they are up-to-date and include recent studies would further strengthen the article's credibility.

after the review of the article, these are the main points that the authors need to consider:

1. Sample Size: The study's small sample size may limit the generalizability of its findings.

2. Demographic Representation: There appears to be a lack of diversity in the participant demographics, which could affect the applicability of the findings to broader populations.

3. Control Group Activities: The control group's activities are not clearly defined, which could impact the study's validity in contrasting the effects of healing garden activities.

4. Duration of the Program: The program's duration may not be sufficient to observe long-term effects or sustainability of the benefits.

5. Quantitative Focus: The study heavily relies on quantitative measures, possibly overlooking qualitative aspects of participants' experiences.

6. Seasonal Variability: The study does not account for the potential impact of seasonal changes on the effectiveness of healing garden activities.

7. Specificity of Activities: The exact nature and intensity of the garden activities are not thoroughly detailed, which can influence the replicability of the study.

8. Psychological Measures: The study could benefit from incorporating a broader range of psychological assessments to capture various dimensions of stress and well-being.

9. Long-Term Follow-up: The study lacks a follow-up to assess the long-term effects of the intervention on participants.

10. Potential Confounders: The study does not sufficiently address potential confounding variables that could influence the outcomes, such as participants' prior exposure to gardening or nature-related activities.

Round 2

Reviewer 1 Report

Comments and Suggestions for Authors

Although improvements are seen overall, there are some problems. Authors should clear these problems.

Introduction

Page 1 line 39-44

In the short term, it disrupts sleep, increases feelings of loneliness and depression, increases heart rate and blood pressure, reduces the body's ability to produce antibodies, weakens the immune system, and slows the recovery of wounds. Stress is not only harmful, but it also has negative health effects, such as heart disease, cancer, diabetes, and depression, and causes drinking, smoking, irregular eating, and irregular exercise habits [2].

These sentences are not organized. What is difference between “harmful” and “negative health effect”? “Depression” is seen twice in these sentences. There is a mixture of diseases and poor health habits. These sentences should be organized.

Page 2 line 74-78

By introducing a garden activity program that increases physical activity and provides psychological and emotional stability to the elderly, the stress of the elderly is positively reduced, and if it is confirmed that the body changes to a healthy state, garden activity can be used as a stress management therapy for the elderly.

This sentence is too long and hard to read because of poor construction.

What does “the body changes to a healthy state” mean Do you mean “improve someone’s health”?

Materials & Methods

2.2 Experimental Procedure

There is not information about how participants were divided and allocated into the experiment group and control group. If this study is RCT, participants were randomly allocated into the two groups.

Line 104 “8 participants each who had a low program participation rate

In general, participant rate means the rate of subjects who participate in someone. “rate of absence” is suitable.

Explanation of “Total power” is insufficient. It is written that “Total Power shows the sum of HF, LF, and VLF waves generated by your heart. Lower values usually mean that you are too tired or sick. Higher Total Power is better, but it should not be too high” in the below link.

Frequency Domain Scores of your Heart Rate Variability | Welltory Help Center

Authors should cite references that can be searched by PubMed or major scientific article data bases. For Example, “15.  Lee, S.D.; Kim, J.Y.; Kim, K.T.; Beon, M.G.; Kim, S.H.; Park, D.I. The study on the activity of autonomic nerve system by using HRV on neurosis. J Int Korean Med fal (1) 2016, 1-6.” Measurement of Total Power is important. It is necessary that researchers can search the reference easily.

Results

Page6 line 170-Page 7 line 184

Authors should change in a form of comparing “Experiment group” and “Control group”. Figures written in these sentences are present in Table 2. In other word, researchers can find such information by watching Table 2 without reading these sentences. I pointed out this problem in the previous review.

Table 4 How is “Stress Level” measured? There is no description.

References

Overall, there are too many references from local journals. Authors should cite articles from international journals that are available.

Comments on the Quality of English Language

Quality of English language was improved. However, some expressions that are wrong are still left.

Reviewer 2 Report

Comments and Suggestions for Authors

Thank you for your thorough response and the modifications made to your manuscript based on the review comments. Your efforts to address each point—such as enhancing the demographic questionnaire, clarifying statistical analyses, and discussing the study's limitations—are commendable and significantly strengthen your paper. The additional data and graphs provide valuable insights, enhancing the clarity and depth of your findings. Your acknowledgment of the limitations and future research directions shows a thoughtful consideration of the study's scope and impact. Your commitment to incorporating feedback is appreciated and contributes positively to the scientific community.

Author Response

Thank you.

If there are no further corrections, please sign the review report.

Round 3

Reviewer 1 Report

Comments and Suggestions for Authors

Improvements are found and further improvements are needed.

Line39-44

It is strange to divide health problem from stress into “in short term” and “in long term”. Depression, feeling loneliness often continue long term. In the reference you cited, the classification of “in the short term” and “in the long term” is not found. It is better to classify health habits, physical problems and psychological problems.

Line 171-185

Authors cannot understand what I pointed out. You should compare the experimental group and control group in the sentences instead of enumerating n (%).

I showed an example of description of demographic characteristics. Authors should refer to other published articles other than this.

The sexual distribution was same between the experimental group and the control group (men 27.3%). All participants in the control group and almost participants in the experimental group were marred. Regarding the distribution of age, the experimental group has higher percent of participants aged 60-69 years (72.7% v.s. 54.5%) and lower percent of participants aged 70-79 years (27.3% v.s. 45.5%). In terms of educational background, the experimental group has lower percentage of participants with high school graduates than the control group (45.5% v.s. 81.8%).

I think “>3 years” means more than 3 years. Authors should check all of this manuscript.

If you cannot describe accurate principle for measuring, you should delete this indicator.

International journals should be cited instead of local journals.

These sentences should be referenced by international journals.

It threatens an individual's quality of life and increases the likelihood of health problems in the human body [3].

3.   Jung, J.Y.; Lee, Y.C.; Lee, H.J.; Cho, W.K. Effect of horticultural therapy on physical activity and stress for elderly. J. People Plants Environ. Sym. Konkuk University, Seoul, 2016; 1, 191-192(Abstr).

For older people, the stress experienced in daily life has a stronger emotional and psychological impact than major life events [5].

5.   Park, M.O.; Seo, H.S.; Lee, S.J.; Koo, B.H. Effect of garden healing activities on stress changes in the elderly. J. Korea Institute of Garden Design 2021, 7(4), 323-334.

There are many articles in international journals with content that meets your sentences.

It is inadequate to reference this article because this sentence says “entire life” but this article is limited to school children.

This stress is not limited to a certain period of development, but is experienced throughout one's entire life, and the type and intensity vary depending on the life cycle or individual [4].

4.   Lee, S.M.; Seo, H.S.; Yoon. H.K.; Jung, Y.B.; Hong, I.K.; Lee, S. The effect of horticultural therapy program based on basic psychological need theory using school garden for stress and school life satisfaction of middle school students. J. Korean Pract Arts Educ. 2020, 26, 181-200.

Comments on the Quality of English Language

As a results of a general survey of leftists

In general, "as a result " does not come in this case.

What does leftists mean?

Round 4

Reviewer 1 Report

Comments and Suggestions for Authors

One problem is still remaining. No description is seen for an indicator "Stress Level" in Table 2 and 4.

Comments on the Quality of English Language

Overall improved.
